# Comparing Human Versus Machine-Driven Cadastral Boundary Feature Extraction

**Emmanuel Nyandwi [1],\*, Mila Koeva [2]** **, Divyani Kohli [2] and Rohan Bennett [3]**

[1] Department of Geography and Urban Planning, School of Architecture and Built Environment (SABE), College of Science and Technology (CST), University of Rwanda, Kigali City B.P 3900, Rwanda

[2] Faculty of Geo-Information Science and Earth Observation (ITC), University of Twente, 7500 AE Enschede, The Netherlands

[3] Department of Business Technology and Entrepreneurship, Swinburne Business School, BA1231 Hawthorn campus, Melbourne, VIC 3122, Australia

\* Correspondence: enyandwi7@gmail.com

**Abstract:** The objective to fast-track the mapping and registration of large numbers of unrecorded land rights globally has led to the experimental application of Artificial Intelligence in the domain of land administration, and specifically the application of automated visual cognition techniques for cadastral mapping tasks. In this research, we applied and compared the ability of rule-based systems within Object-Based Image Analysis (OBIA), as opposed to human analysis, to extract visible cadastral boundaries from very high-resolution World View-2 images, in both rural and urban settings. From our experiments, machine-based techniques were able to automatically delineate a good proportion of rural parcels with explicit polygons where the correctness of the automatically extracted boundaries was 47.4% against 74.24% for humans and the completeness of 45% for the machine compared to 70.4% for humans. On the contrary, in the urban area, automatic results were counterintuitive: even though urban plots and buildings are clearly marked with visible features such as fences, roads and tacitly perceptible to eyes, automation resulted in geometrically and topologically poorly structured data. Thus, these could neither be geometrically compared with human digitisation, nor actual cadastral data from the field. The results of this study provide an updated snapshot with regards to the performance of contemporary machine-driven feature extraction techniques compared to conventional manual digitising. In our methodology, using an iterative approach of segmentation and classification, we demonstrated how to overcome the weaknesses of having undesirable segments due to intra-parcel and inter-parcel variability, when using segmentation approaches for cadastral feature delineation. We also demonstrated how we can easily implement a geometric comparison framework within the Esri's ArcGIS software environment and firmly believe the developed methodology can be reproduced.

**Keywords:** cadastral intelligence; manual digitisation; expert parameterisation; land administration; land management; automatic feature extraction; Object-Based Image Analysis

---

## 1. Introduction

The emergence of artificial intelligence (AI) concepts, methods and techniques ushered in a new era of the longstanding philosophical debate and technical competition between the merits of 'human' versus 'machine'. Machine-based techniques exhibit computation capabilities capable of handling complex issues not quickly solved by humans [1]. Meanwhile, as people become more intelligent, they can prescribe precision and program performance of a high quantity task to a machine, such as the extraction of cadastral features from images. Considering that only ~30% of land ownership units

worldwide are captured in formal cadastres and land registration systems [2,3], automation techniques could be a supportive tool for the generation of digital parcel boundaries, enabling faster registration and mapping of land rights.

In light of the recent enhancement to machine-driven feature extraction techniques, the current study aims to measure the ability of machine-based image analysis algorithms, against manual digitising, in extracting cadastral parcel boundaries from very high-resolution remotely sensed images.

## 1.1. Cadastral Intelligence

Our definition of cadastral intelligence draws on the 1983 Howard Gardner's theory of multiple intelligences in the area of spatial intelligence [4–6]. Gardner defines spatial intelligence as the ability to perceive the visual–spatial world [4], to localise and visualise geographic objects [5]. From a remote sensing perspective, spatial intelligence ranges from visually discriminating geographic objects using reasoning, and then drawing and manipulating an image [6]. In the cadastral domain, the ability to acquire and apply spatial intelligence in detecting cadastral boundaries is referred to as cadastral intelligence [7,8].

Recently, developments in artificial intelligence have reshaped spatial intelligence into "automated spatial-intelligence" [9,10]. From an artificial intelligence perspective, spatial-intelligence is constituted by the procedural knowledge exhibited through computational functions represented by a set of rules and structural knowledge which allow the establishment of the relationship between image-objects and real-world geographical objects [11]. Artificial intelligence has implied that many algorithmically-trained perception-capable computing models exist, that beside human operators, can perceive and understand geographic data and recognise geographically referenced physical features [10]. In the contemporary era, we can witness substantial progress in remote sensing where automatic image registration allows for the handling of huge volume of remote sensing images [12] efficiently [13,14]. Regarding feature extraction from remotely sensed images, automation, though difficult to configure and implement, is the eventual solution to the limitations of manual digitisation [15], and this is with no exception in cadastral mapping [7,8]. For these reasons, the use of artificial intelligence and automation is generally gaining traction within geoinformatics and land administration research domain [16].

## 1.2. The Quest of Automation in Cadastral Mapping

The cadastre is a foundation for land management and development [17–21]. An appropriate cadastral system supports securing property rights and mobilising land capital, and without it, many development goals for countries are either not met, or greatly impeded [19]. A major issue, however, is that only around one-third of land ownership units worldwide are covered with the formal cadastre [2,3]. The full coverage of cadastre is arguably impeded, in part, by procedural and costly conventional surveying approaches. The latter suggests that all cadastral boundaries must be 'walked to be mapped' [22,23] making it resource intensive. Surveying is thus the costliest process when registering landed or immovable property [24], incurring 30–60% of the total cost of any land registration project [24,25]. The consequence is a growing aversion towards land registration, lest the benefits of it would not compensate for the money spent [26].

Emerging geospatial technologies have made it possible to democratise mapping and registration activities, conventionally undertaken by highly qualified, if not expensive, surveying experts. Mobile devices equipped with simple web mapping apps, incorporating the ever increasing amount of high quality aerial imagery and connected to the cloud, have seen a rise in the 'barefoot' (Also referred to as 'grassroots surveyor' and 'community mapper') surveyor and more recently the 'air-foot' surveyor [2,7], in the context of unmanned aerial vehicles (UAVs) applications. The latter, by substituting for the use of ropes, groma, tape measure, theodolite, total stations and walking outside in the field [27], allows the detection of visible cadastral boundaries based on their patterns with respect to appearance and form from a distance [28,29]. Owing to advances in remote sensing, image-based cadastral demarcation approaches have been proposed and have been experimented with in countries including Rwanda,

Ethiopia and Namibia. Experimentation demonstrates the effectiveness of remote sensing image-based demarcation in delivering fast-track land registration [30].

Meanwhile, recent developments in the field of computer vision and artificial intelligence have led to a renewed interest in cadastral mapping where machine algorithms, able to mimic humans in exhibiting spatial intelligence, could potentially be used to automate boundary extraction. The latter approach could allow tapping existing opportunities with very high-resolution remote sensing data, from various sources and wide coverage. Contemporary satellites, besides manned airborne photography, have offered sub-metre spatial resolution images, since the late 1990s [31,32]. Furthermore, with UAV, it is now possible to acquire centimetre-level image resolution and point cloud data, allowing operators to uncover features occluded by vegetation [29,33,34].

When compared to manual on-screen digitisation, automation offers many potential advantages, including the removal of inconsistency errors resulting from different users performing digitisation. Automation also allows coverage of wide areas with minimum labour, and supports cheap and up-to-date fit-for-purpose solutions that aim to target existing societal needs [35,36]. Automation could assist with the massive generation of digital property boundaries. Despite the two-sorted ontology of boundaries, implying that some of the property boundaries are invisible [37,38], the majority of cadastral boundaries are believed to be self-defining and can be extracted visually [39]. Visible boundaries are marked by extractable physical features like fences, hedges, roads, footpaths, trees, water drainages, building walls and pavements [40,41] which are detectable using remote sensors. Therefore, these features can be extracted from remote sensing data to generate boundaries with the features they represent [28]. Such features of cadastral boundaries can be detected based on their specific properties like being regular, linear-shaped or with limited curvature in their geometry, topology, size and spectral radiometry or texture [42].

Machine-based image analysis approaches, applicable for automated cadastral boundaries extraction, can be grouped into two categories: (i) pixel-based approaches (PBA) and (ii) object-based approaches, also well known as Object-Based Image Analysis (OBIA) [28]. The first approach only considers spectral value or one aspect for boundary class [43]. Thus, PBA algorithms, with the exception of state-of-the-art convolution neural network (CNN) [28,44–46], may result in a "salt and pepper" map when applied to very high-resolution images [47]. Due to the lack of an explicit object topology that might lead to inferior results compared to those from the human vision, PBA falls short of expectations in topographic mapping applications [11]. Unlike PBA, objects resulting from OBIA are features with explicit topology, meaning they have geometric properties, such as shape and size [48]. This makes OBIA suitable for extracting cadastral boundaries [28,39,49].

In brief, it appears there are many potentialities for the automation of feature delineation using artificial intelligence tools such as state of the art CNN models and OBIA. However, the major problem remains: how to make automation an operational solution [50], especially in cadastral mapping, where properties need to be delineated with high precision specification geometrically and topologically. This infers the presence of a compelling need to research more on the usability and applicability of AI-based cadastral intelligence in land administration [16]. Therefore, our study is built on the necessity to explore the potentialities of machine-based image analysis algorithms to extract cadastral parcels. While, theoretically some automation tools could even outperform human operators to extract features from images [11], little is said on their performance compared to humans within a cadastral domain-specific application. The focus of this research is, therefore, to investigate the extent to which automatically captured cadastral boundaries align with existing geometric standards of cadastres.

## 2. Materials and Methods

This study applied a comparative approach using two case sites, an urban setting and a rural setting within Kigali city, Rwanda (Figure 1), globally recognised to be among first countries where image-based demarcation was applied to build a nationwide cadastre system at a low cost. The sites were selected based on the availability of very high-resolution satellite images, the hypothesised visual

detectability of cadastral boundaries, and convenience of accessing reference datasets for comparison. The study used pan-sharpened images for extracting rural parcels and urban plots with building outlines. The study involved three main steps: pre-processing of the images, boundary extraction (automation and human digitisation) and geometric comparison.

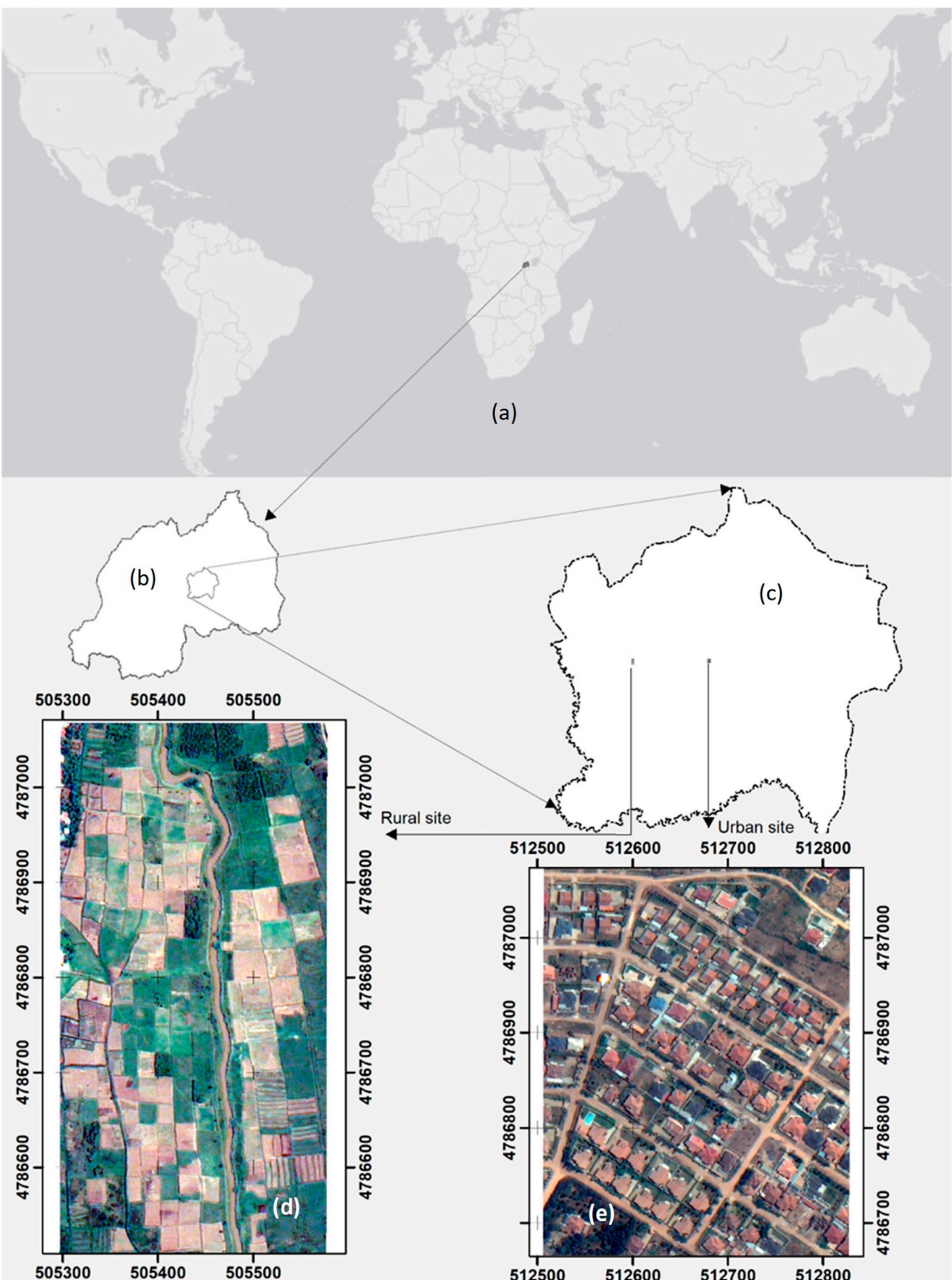

**Figure 1.** Case study location: (**a**–**e**) show, respectively, the study area on global scale; Rwanda; the city of Kigali; the rural site and urban site. Coordinates system is ITRF2005.

*2.1. Pre-Processing*

Pre-processing concerned preliminary operations: (1) sub-setting the image to eliminate extraneous data and constrain the image to a manageable area of interest and (2) pan-sharpening by fusing the 2 m resolution multispectral with 0.5 m-resolution panchromatic World View-2 satellite images, tiles of $280 \times 560$ m and $320 \times 400$ m, respectively, for enhanced visual interpretability and analysis. For pan-sharpening, the nearest-neighbour diffusion algorithms available within the Environment for Visualising Images (ENVI) software, was applied owing to its advantage of enhancing the salient spatial features while preserving spectral fidelity [51].

*2.2. Parcels and Building Outline Extraction*

2.2.1. Automatic process

For automation, we applied the OBIA approach. OBIA can combine spectra, texture, geometry and contextual information to delineate objects with explicit topology, shape and size [48], which are key aspects of the cadastral index map. In our study, we tested both fully automated parameterisation using the estimate scale parameter (ESP2) tool [52] and expert knowledge parameterisation using the Trimble's eCognition software which provides a development environment for object-based image analysis [53]. The extraction of parcels and building outlines were based on object characteristics. Building outlines, in general, are missing in the Rwanda's cadastral database [2], while they are essential for property taxes correction [53]. Thus, automatic extraction of building outlines with correct shapes could be useful in updating the legal cadastral database.

Fully Automated Parameterisation

Initially, the automated estimate scale parameter (ESP2) tool was applied to automate parcels and building outlines extraction. The appeal of this tool is that it supports automated optimisation of the scale parameter (SP), which is the key control in multiresolution segmentation (MRS) process and it is fully automatic. It produces segments at three spatial levels representing different scales, based on the concept of local variance with one push of the button.

Expert Knowledge for Parameterisation

While the ESP2 tool can produce segments with one push of the button, it has its limit when dealing with parcel boundaries with contextual morphological variability, which necessitates expert contextual knowledge for their extraction. In fact, the selection of parameters such as scale parameter in segmentation is an objective decision [54] requiring reasoning of the user who can instruct the machine. In this context, our study mainly employs expert user-developed rule sets for automated extraction of cadastral features.

In the rural area, parcel automation involved three main operations: segmentation, classification and boundary enhancement, and used different techniques such as chessboard and multiresolution segmentation. Due to intra-parcel and inter-parcel variability, segmentation may generate undesired segments [50]. To deal with this issue, an iterative approach of segmentation and subsequent classification was applied, and we created several classes of parcels and then a merged an output containing all parcels.

As indicated in Figure 2, the target input area (see the size of input image) became smaller as we segmented and classified image features into parcels and merged unclassified image features, as input for the next round of segmentation and classification and so forth.

For segmentation, the chessboard segmentation with a value of five was used to generate image objects of $5 \times 5$ pixels. The open source map (OSM) data set—river for rural site and roads for urban site—was used as thematic layers in chessboard segmentation. Based on expert ground knowledge, the distance to vector contextual information was used to classify image objects within 10-m from the river as the class river strip. Then, MRS was applied to extract ditches. To enhance the detectability of

ditches, the texture after Haralick—derived from the grey level co-occurrence matrix (GLCM), best known for localising texture boundaries [55]—was used as a temporary image layer. After extracting river strips and ditches, the extraction of parcels with varying size, shapes involved tuning the scale parameters for segmentation and geometric indexes for classification. It is worth noting that unlike in ESP2, manual scale parameterisation was a trial and error tuning process, and used selected parameters (see Table 1). In addition, it must be clearly understood that the classification (Table 1) was not intended to have different thematic classes, i.e., parcels 1–9 are not different thematic classes, but features with different size/shape/texture that belong to the same "Parcel" class.

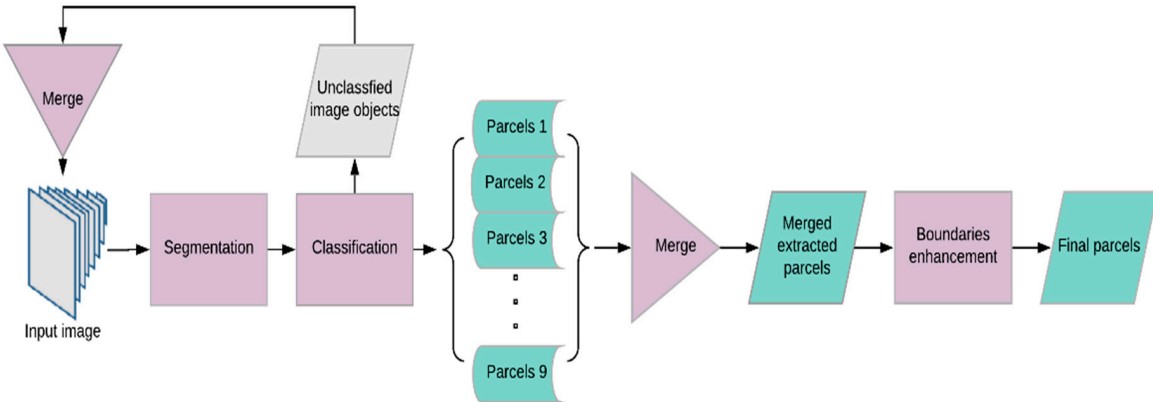

**Figure 2.** Automated rural parcels' extraction workflow.

**Table 1.** Parameter tuning with different steps followed in rule sets.

| Rural | | | |
|---|---|---|---|
| | **Operation** | | **Parameters** |
| Removing non parcel features | | Chessboard segmentation. | Size = 5 pixels; |
| | | Contextual information | Distance to river = 11 pixels, distance to drainage = 5 pixels |
| | | MRS segmentation | Scale = 10; shape = 0.1; compactness = 0.5 |
| | | GLCM entropy features | (Quick 8/11) R, (all directions) |
| | | MRS segmentation | Scale = 20; shape = 0.4; compactness = 0.8 |
| | | Classification | Ditches: elliptic fit = 0; Asymmetry = 0.92 |
| Parcels extract-ion | Iteration 1 | MRS segmentation | Scale = 70; shape = 0.5; compactness = 0.9 |
| | | Classification | Parcels-1 (shape index < 1.2 and rectangular fit ≥ 0.88 and area >300 $m^2$ |
| | Iteration 2 | MRS segmentation | Scale = 70; shape = 0.6; compactness = 0.8 |
| | | Classification | Parcels-2 (shape index < 1.3 and rectangular fit ≥ 0.9 and area ≥ 200 $m^2$ |
| | Iteration 3 | MRS segmentation | Scale = 35; shape = 0.4; compactness = 0.8 |
| | | Classification | Parcels-3: rectangular fit ≥ 0.93 |
| | | Classification | Parcels-4: shape index = 1.345 and rectangular fit ≥ 0.88 and area ≥ 425 $m^2$ |
| | Iteration 4 | MRS segmentation | scale = 35; shape = 0.5; compactness = 0.8 |
| | | Classification | Parcel 5 = shape index ≤ 1.35 and rectangular fit > 0.9 and area >= 400 $m^2$ |
| | Iteration 5 | MRS segmentation | Scale = 70; shape = 0.5; compactness = 0.9 |
| | | Classification | Parcel 6 = shape index ≤ 1.4 and area ≥ 360 $m^2$ |
| | Iteration 6 | MRS segmentation | Scale = 60; shape = 0.5; compactness = 0.8 |
| | | Classification | Parcel 7 = shape index ≤ 1.4 and rectangular fit > 0.9 and area > 400 $m^2$ |

**Table 1.** *Cont*.

| | | |
|---|---|---|
| Iteration 7 | MRS segmentation Classification | Scale = 70; shape = 0.6; compactness = 0.9 Parcels 8 = shape index ≤ 1.4 and rectangular fit > 0.85 |
| Iteration 8 | MRS segmentation Classification | Scale = 90; shape = 0.5; compactness = 0.8 Parcels 9: density ≥ 1.6 |
| Enhancemen-t | Opening operator Chessboard Growing region | Size 1 × 1 pixel, Loop: parcels <unclassified> = 0 |
| **Urban** **Buildings extraction** | | |
| Removing road strips | Chessboard segment. Contextual information | Size = 1 × 1 pixel, Distance to OSM road set = 7 m |
| Removing vegetation | Classification | NDVI > 0.73 Maximum difference < 2.05 |
| Buildings | MRS segmentation Classification | Scale = 70; shape = 0.8; compactness = 0.9 Area > 150 m$^2$ |
| **Fences/parcels extraction** | | |
| Contrast segmentation/Edge ration splitting on blue band | | Chessboard tile = 30; minimum threshold = 0; maximum threshold = 250, step size = 50 |

In the urban area, automated extraction of building outlines used chessboard segmentation to split the image into equal smaller objects of 0.5 m × 0.5 m and buffers of 7 m from the central line of roads to eliminate road strips. To obtain building shapes, the normalised difference vegetation index and maximum difference features were used to separate vegetation (garden) and other non-roof objects from roof objects. Contrast segmentation was used to automate the extraction of parcels that were marked by dark strips.

### 2.2.2. Manual Digitisation

Five human cadastral experts were tasked to individually hypothesise and manually digitise cadastral boundaries using the same dataset as used for automation. Using more than one person allowed to assess human consistency. In accordance with [56], experts were identified based on cadastral domain professional qualifications, experience and memberships of the recognised surveying professional body (Table 2). The domain experts had extensive knowledge and expertise. They were familiar with the subject at hand and understood analyses by automatic, abstract, intuitive, tacit and reflexive reasoning. They could perceive systems, organise and interpret information [57]. The team of experts was provided with an extraction guide that clearly describes cadastral objects to be extracted, input dataset with clear digitising rules.

**Table 2.** Human operators.

| Expert ID | Qualification | Professional Body | Experience |
|---|---|---|---|
| A | Master in Geoinformation and Earth Observation | National cadastre | 8 years |
| B | Bachelor of Science in Geography | National cadastre | 8 years |
| C | Bachelor of Science in Geography | National cadastre | 8 years |
| D | Bachelor of Science in Land Surveying | Organisation of surveyor | 5 years |
| E | Bachelor of Science in Geography | National cadastre | 5 years |

### 2.2.3. Geometric Comparison of Automation versus Humans

In our study, geometric precision, that is usually more important than the thematic accuracy of spatial feature delineation [39], was considered the key aspect in measuring the machine's performance against human operators, for parcels and building outlines extraction. Considering that an important

aspect of developing systems for automated cartographic feature extraction is the rigorous evaluation of performance, based on precisely defined characteristics [58], we decided to use accurate reference boundaries, measured out in the field rather than current legally recognised boundaries. Field surveys used precision survey tablets running Zeno field mapping software. Access to Rwanda GNSS-CORS-RTK (Global Navigation Satellite System-Continuously Operating Reference Stations (CORS). Real-Time Kinematic) differential corrections allowed centimetre level accuracy for reference data. While existing legal boundaries are not accurate enough (with an estimate shift of 1 to 5 m of the 'true' position [59]) to not serve as reference data, they were leveraged on to establish precise boundaries during field survey.

Considering the scenarios elaborated (Figure 3), we can compute geometric errors such as over-segmentation error (OS), under-segmentation error (US), edge error ($ED_{err}$), fragmentation error or Number-of-Segments Ratio (NSR) and shape error ($SH_{err}$) that evaluate the degree of mismatching between the reference cadastral object and the corresponding extracted cadastral object on the map as in [60,61] and as follows:

$$OS_{err}(R_i, C_i) = 1 - \frac{R_i \cap C_i}{R_i} \tag{1}$$

$$US_{err}(R_i, C_i) = 1 - \frac{C_i \cap R_i}{C_i} \tag{2}$$

$$NSR = \frac{abs(N_r - N_c)}{N_r} \tag{3}$$

where, $N_r$ is the number of polygons in the reference dataset and $N_c$ the number of corresponding extracted parcels.

$$ED_{err}(R_i, C_i) = 1 - \frac{\varepsilon(R_i) \cap (C_i)}{\varepsilon C_i} \tag{4}$$

where $\varepsilon(R_i)$ denotes a tolerance introduced to extracts the set of edge area from a generic region $R_i$ in the recognition of parcel borders. In our study, to find out the tolerance distance that should be used for assessing the discrepancy between reference boundary lines and extracted boundary lines, the base image was overlaid with the surveyed reference data. The shift of boundary lines, ditches on image from surveyed lines, could be measured using the measuring tool in ArcGIS. A shift of 0 to 4 m was identified. Therefore, $\varepsilon(R_i)$ took a value of 4 m.

$$SH_{err} = \|sf(R_i) - sf(C_i)\| \tag{5}$$

where, a shape factor $sf$ could be one of several geometry indices like asymmetry, border index, compactness, density, elliptic fit, main direction, radius of largest enclosed ellipse, rectangular fit, roundness or shape index [61], the latter was used for ease of computation. It is calculated from the border length of the object divided by four times the square root of its area:

$$Sf = \frac{Perimeter}{4\sqrt{Area}} \tag{6}$$

Knowing the geometrical error of the individual classified objects ($Err_i$), a global geometric error ($Err_n$) for n extracted parcels can computed as:

$$Err_n = \frac{1}{n}\sum_{i=1}^{n}(Err_i) \tag{7}$$

In an ideal case, for Equations (1), (2), (4)–(6), the optimum value is 0 and 1 as the worse performance. Furthermore, a one-to-one relationship is obtained where desirably one parcel in the reference is explained by one parcel in the extracted data set (Figure 3a–c, e). In other cases of one-to-many or many-to-one correspondence (Figure 3d), omission errors and commission errors called

false negatives and false positives metrics were determined. The false positives (FP) are parcels, which were erroneously included by either machine or human experts The false negatives (FN) are parcels not detected by either human or machine but they exist in the reference dataset. The performance of machine versus humans could be also estimated as the portion of extracted parcels that could match their corresponding references (correctness) or the portion of reference parcels that could be reproduced by extraction (completeness) [28].

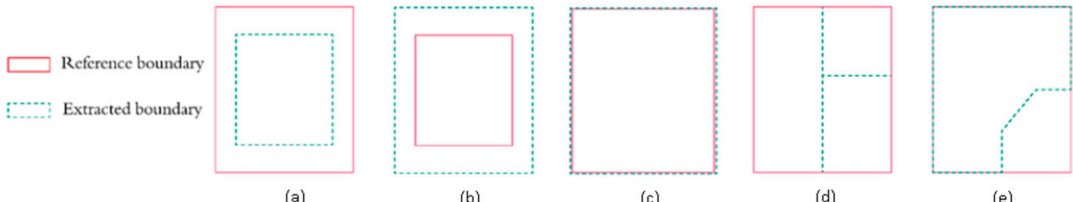

**Figure 3.** Geometric error scenarios: (**a**) over segmentation; (**b**) under segmentation; (**c**) edge error; (**d**) fragmentation error and (**e**) shape error.

In (a), the areas that fall outside the green zone are called over-segments, i.e., the areas omitted from the reference parcel polygon. (b) The area beyond the red line is under segment and committed to the reference parcel. (c) Edge error, is where boundaries of extracted parcel mismatch boundaries of the reference object. (d) Fragmentation error, is where a classifier has split the parcel into several fragments. (e) The shape in green has deviated from the reference shape in red.

Mathematically,

$$\text{FN (omitted)} + \text{TP (detected)} = \text{Reference,} \tag{8}$$

$$\text{FP (Committed)} + \text{TP (Correctly detected)} = \text{Extracted,} \tag{9}$$

$$\text{Correctness} = \frac{\text{Extracted} \cap \text{Reference}}{\text{Extracted}} \times 100, \tag{10}$$

$$\text{Completeness} = \frac{\text{Reference} \cap \text{Extracted}}{\text{Reference}} \times 100 \tag{11}$$

A framework for comparing extracted and reference parcels was developed in the Esri ArcGIS environment (Figure 4). First, we labelled all parcels with unique identifiers. To allow the comparison of each individual parcel in the reference set with the corresponding parcel in the extraction set, the splitting by attributes tool was used to split the index cadastral map into individual parcels. Then the batch intersect tool was used to calculate the intersection for each parcel in the reference data with each corresponding parcel in the extraction set. Resulting intersects were then merged to have one attribute table containing the areas of intersection of extracted parcels and reference parcels. These values were to be fed into Equations (1)–(4). The shape index was computed using the Esri ArcGIS geoprocessing 'calculate field' tool.

For the edge shifting computation using ArcGIS, a model (Figure 5) was built to automate the process. The edge comparison used the perimeter of parcels. In doing so, we first converted reference and extracted parcels to lines. Then, a buffer, i.e., distance tolerance of 4 m was applied to reference lines, and a clip tool was used to get the length from the total perimeter of extracted parcel, which was correctly extracted to match the reference boundary lines. By clipping automated lines with buffered reference lines, two cases were possible: (1) the extracted parcel fully matched the reference parcel and we could use convert lines to polygon to restore the original extracted polygon and (2) when only few lines belonging to one extracted parcel could match the reference lines. Keeping in mind that in a cadastral map, a line is shared by two parcels, and converting parcels into lines results in a shared line with two different identifiers. To know what portion of the line belongs to which parcel we used dissolve by ID tool.

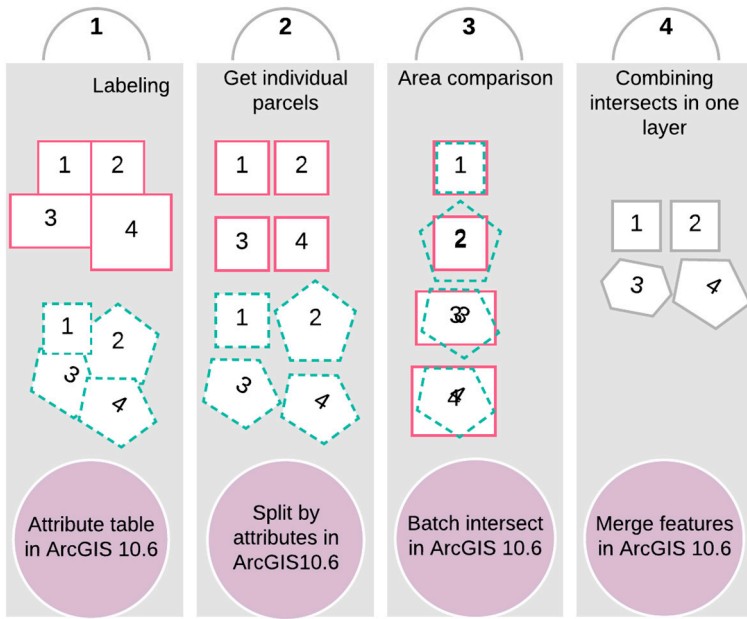

**Figure 4.** Implemented framework for computing geometric discrepancies.

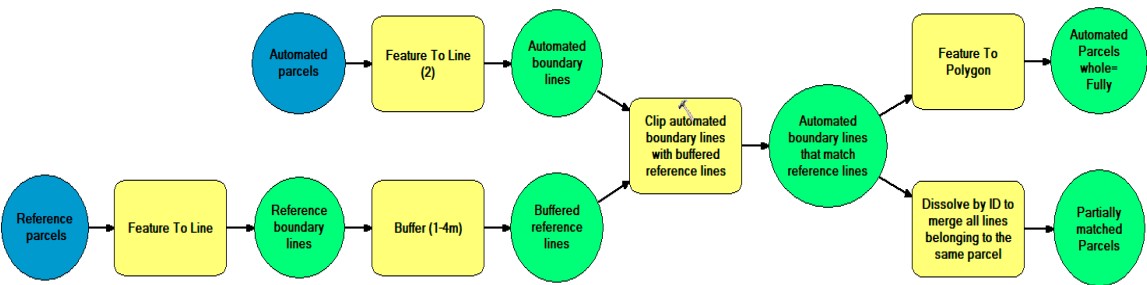

**Figure 5.** Model for assessing edge error between reference and automated boundary lines.

## 3. Experimental Results

This section presents parcel and building extraction results by human and automation/machine means, and comparing the performance of human operators against a machine. Extraction results are disaggregated by sites: rural versus urban.

### 3.1. Extraction of Parcel Boundaries in Rural Area

In the rural area, cadastral professionals hypothesised and digitised parcels following visible features such as ditches according to their expertise. Figure 6a–e below presents manually digitised rural parcels. Figure 6f shows the legal boundaries from the national cadastre.

The automated OBIA approach resulted in ragged and highly inaccurate segments (Figure 7a) when using the fully automated estimate scale parameter (ESP2) tool, with the shape factor of 0.1 and the compactness of 0.5 for segmentation. The results were improved by modifying the shape factor and compactness to 0.5 and 0.8, respectively (Figure 7b). As indicated in Figure 7c, better results were obtained by using the user-developed rule set based on experts' ground knowledge since it was more adapted to context than the ESP2 tool.

During automation, the resulting boundaries had dangling features that do not meet cadastral geometry and topology requirements. In Figure 8a, the parcel in red had a dangling area that needed to be trimmed off, but also ditches—dark strips—that separate parcels need to be represented as line and not as polygons. The morphology operator within eCognition was used to improve the boundaries. In Figure 8b, the pixel-based binary morphology operation is used to trim dangling portion off the main parcels. Morphology setting was done based on instructions from the eCognition reference manual.

In Figure 8c, ditches and other loosely extracted features are sliced into smaller image objects using chessboard segmentation. For smoother results, the object size is set to the smallest size possible, in our case to 1. In Figure 8d, split segments are set to 'merge neighbouring parcels', and parcel boundaries are improved.

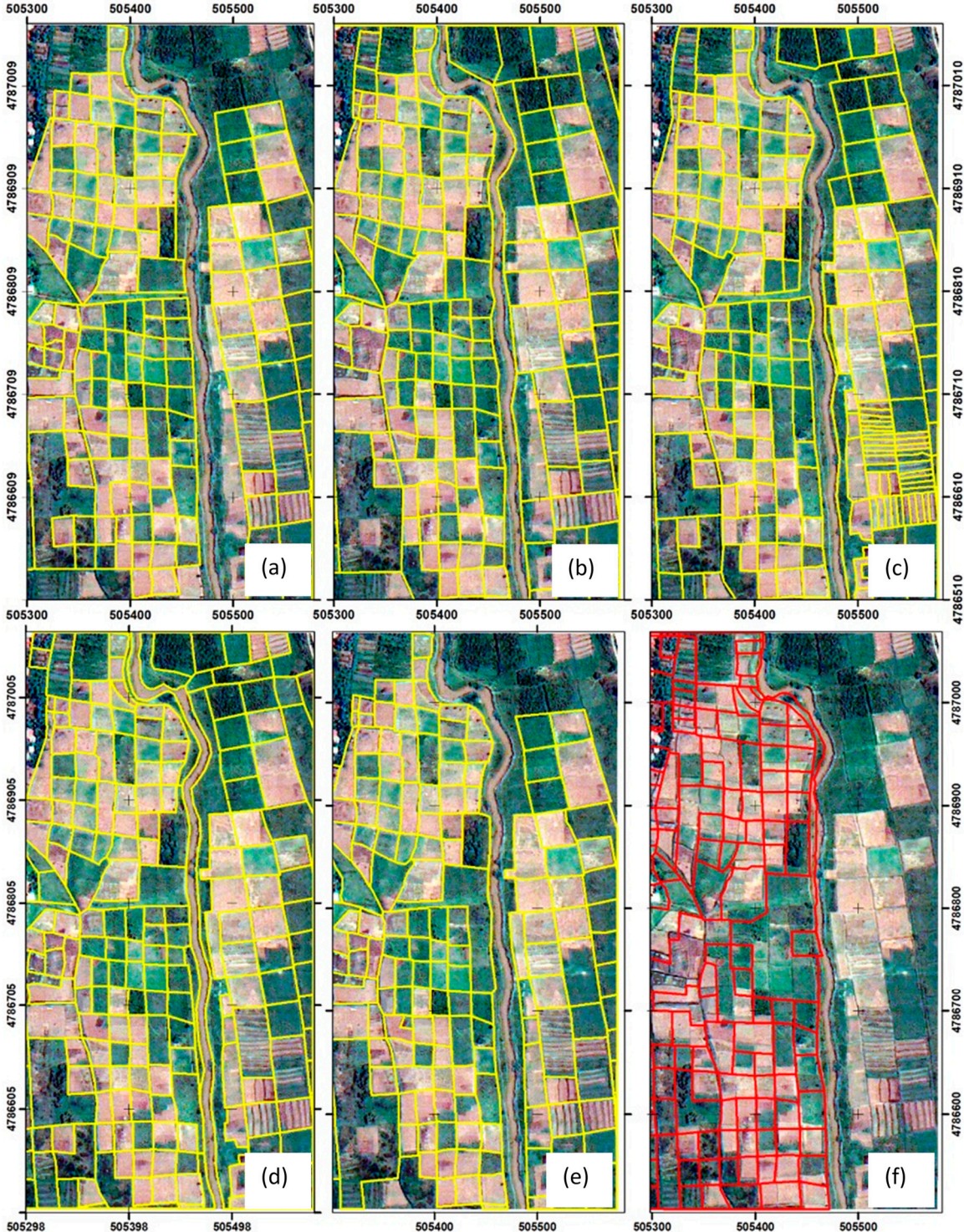

**Figure 6.** Manually digitised parcel boundaries. (**a**–**e**): manually extracted boundaries. (**f**): existing legal boundaries

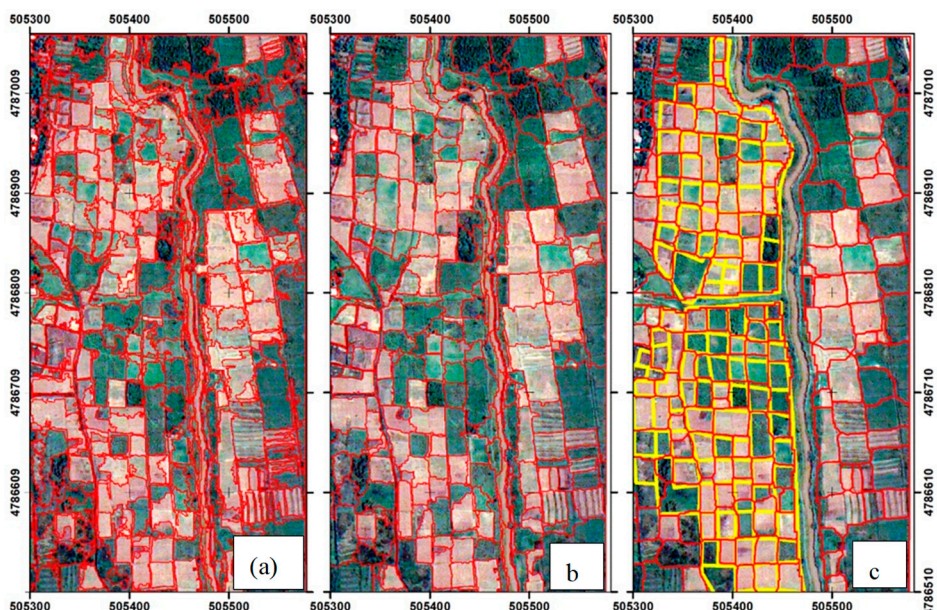

**Figure 7.** Automatically extracted parcels boundaries: (**a**) result of the automated Estimation of Scale Parameter (ESP2) tool intact; (**b**) results of a modified ESP2 and (**c**) results obtained using expert user-developed rule set. Automated parcels (in red) are overlaid with reference parcels (in yellow).

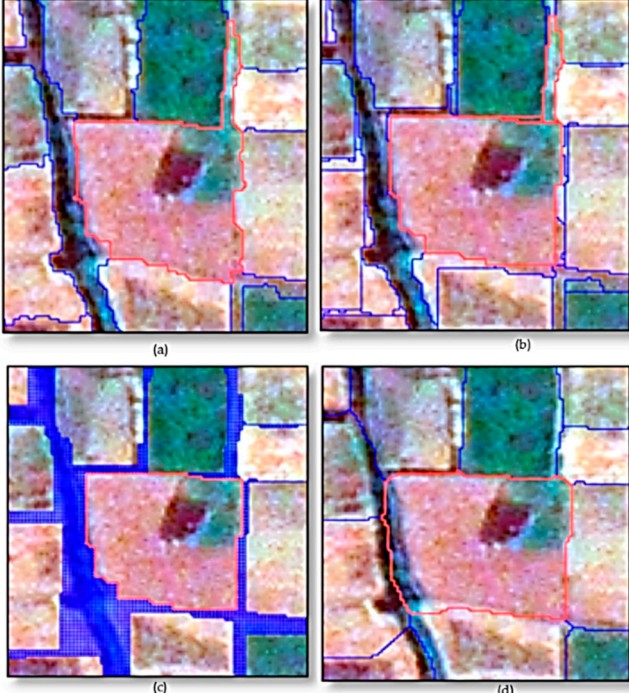

**Figure 8.** Boundary enhancement: (**a**) boundaries before enhancement; (**b**) dangling areas are trimmed off; (**c**) dangling features are split into smaller chunks and (**d**) parcels are dilated to include nearby dangling chunks.

## 3.2. Extraction Parcels in Urban Areas and Building Outlines

The extraction of parcels and buildings in the urban area relied on visible fences, roofs and roads. Figure 9a–e presents reference parcels (in red) overlaid with manually extracted urban plots (in yellow) and automated parcels in Figure 9f.

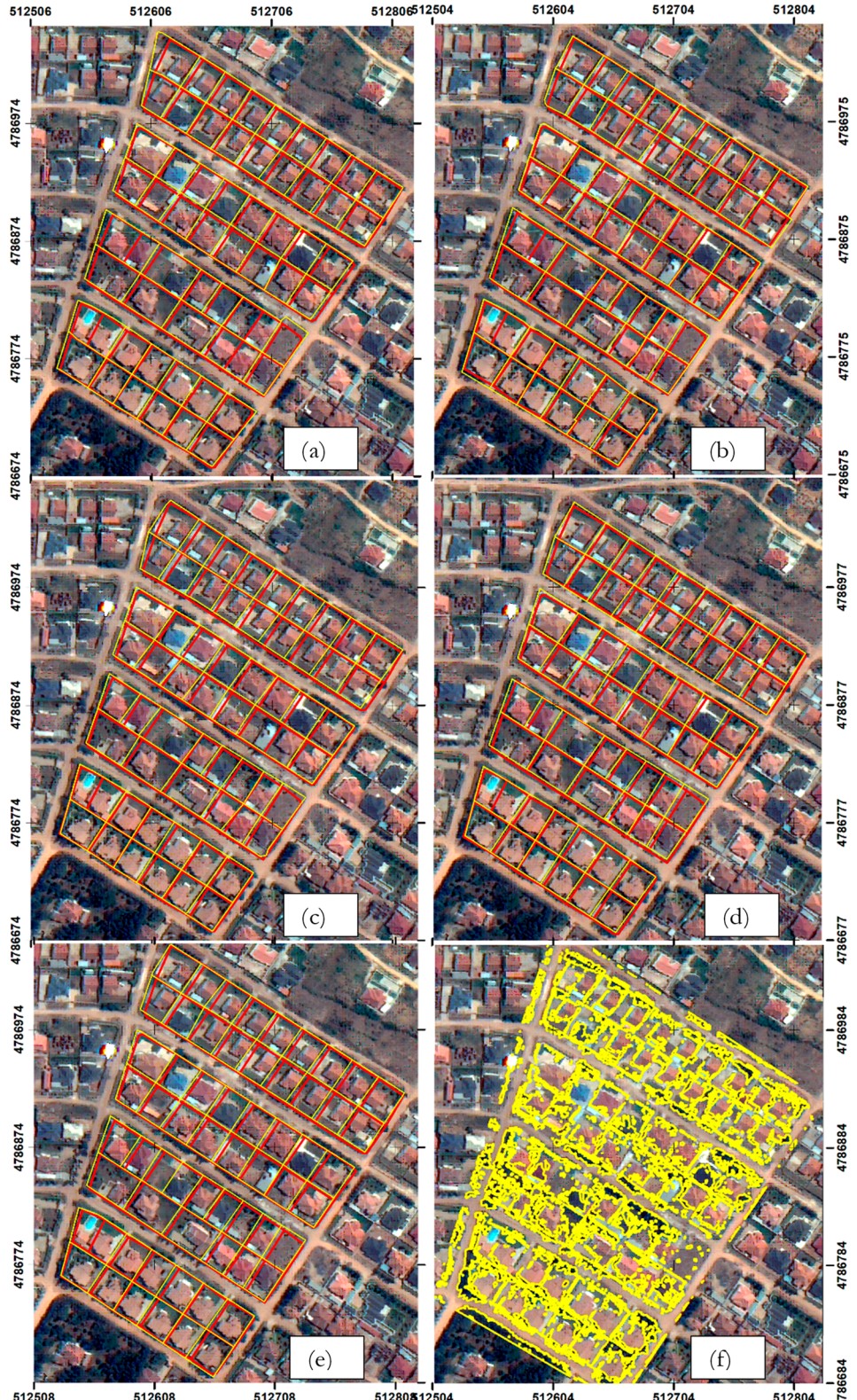

**Figure 9.** Extracted urban parcels. (**a–e**): manually extracted parcels by five experts. (**f**): automated parcels boundaries

In addition to building plots, we applied both automatic and manual techniques for the extraction of building outlines. Figure 10 represents results from the expert team's manual digitisation (Figure 10a–e) and automatically extracted building outlines (Figure 10f).

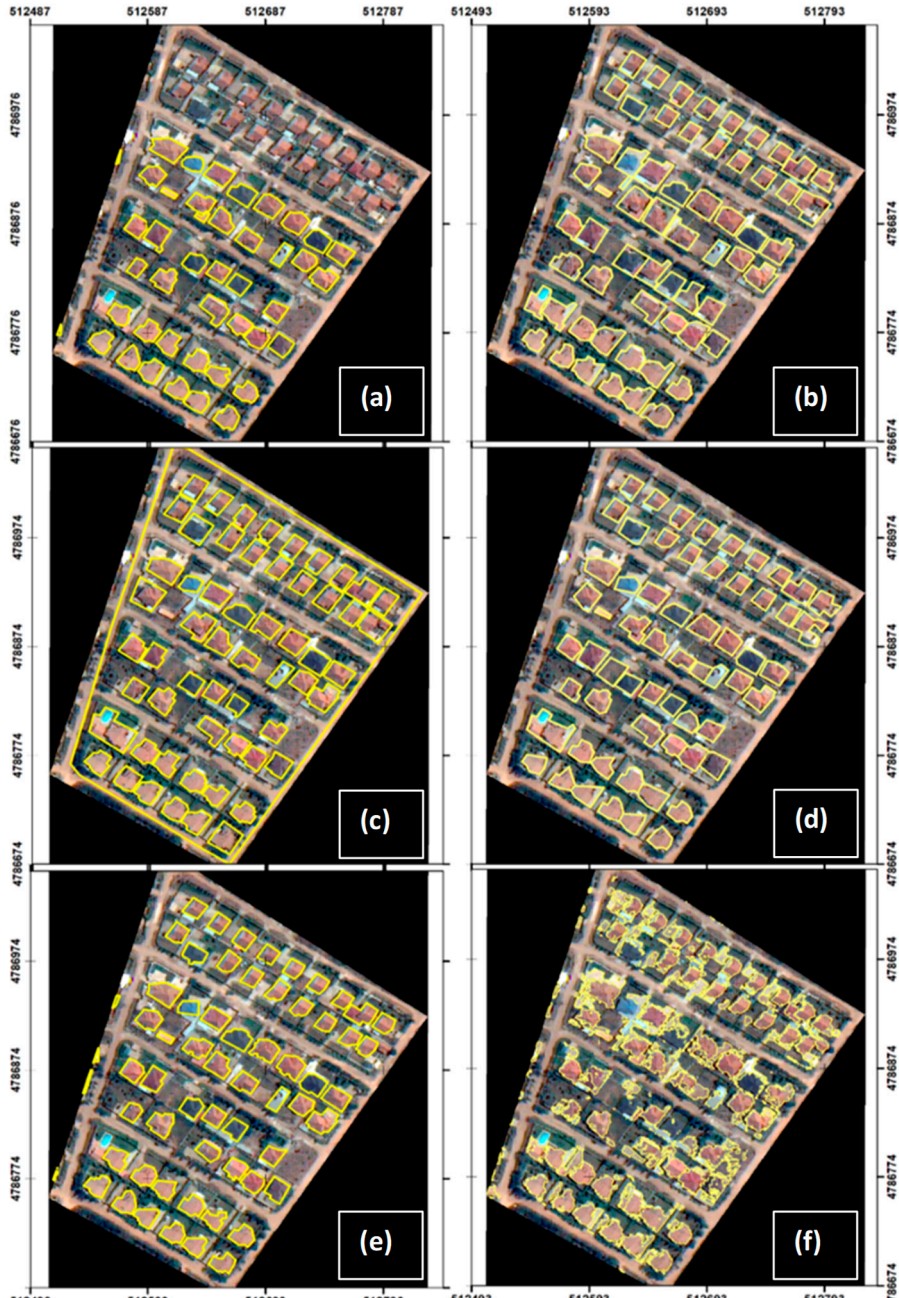

**Figure 10.** Extracted buildings plots in urban area: (**a–e**) manual digitisation boundaries and (**f**) automation results.

The automation output in the urban area was counterintuitive, at least compared to human visual interpretation. The results show that automation resulted in poorly structured parcel boundaries compared to the manually digitised parcels. As shown on Figure 10f, the machine faced difficulties in trimming pavements and tiny structures from the main buildings. Blue and black roofed buildings were omitted as they spectrally appeared to resemble vegetation. On the contrary, humans were more precise and concise.

## 3.3. Geometric Comparison of Automated Against Manually Digitised Boundaries

Geometric discrepancies between each reference parcel and the corresponding extracted parcel were determined by overlaying automatically produced parcels with manually digitised and field surveyed parcels. A distance tolerance buffer of 4 m was applied considering the shift of boundaries

inherent in the source image. Note that the comparison was done only for the rural site, where automation results were geometrically comparable with manual and reference parcel polygons.

In Figure 11, violin graphs are shown, on which white dots mark the median, illustrating the full distribution of discrepancies between reference parcels, automated and manually digitised parcels.

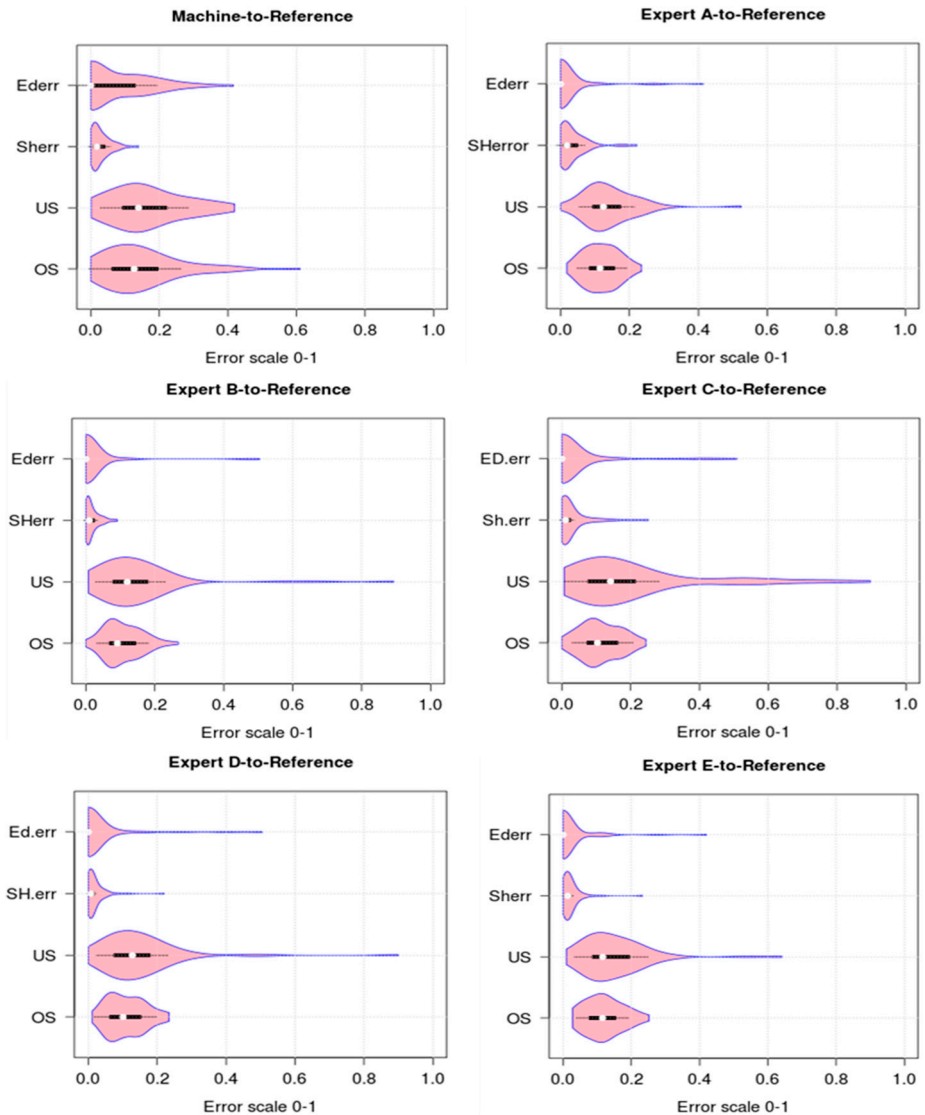

**Figure 11.** Clustering of geometric discrepancies. Ederr: edge error; SHerr: shape error; US: under segmentation and OS: over segmentation.

In Figure 11, the wider section of the violin plot [62] indicates higher numbers of extracted parcels within the given error value, whereas the skinnier section shows the reverse case. The graphs allow us to examine the behaviour during all instances, i.e., the variation and likeness in the full distribution and the pattern of responses for machine and human can be visualised and compared.

The comparison of machine intelligence to expert knowledge was also done by comparing automated parcels with hypothesised and manually digitised parcels. Unlike the previous comparison, the analysis of automated parcels against hypothesised parcels by experts did not necessarily consider the correctness of detection, i.e., the degree of extraction coinciding with cadastral boundaries. The focus is the ability to detect visible boundary features on the image. As it can be observed, Figure 12 shows data in agreement with Figure 11, with respect to the shapes of extracted parcels.

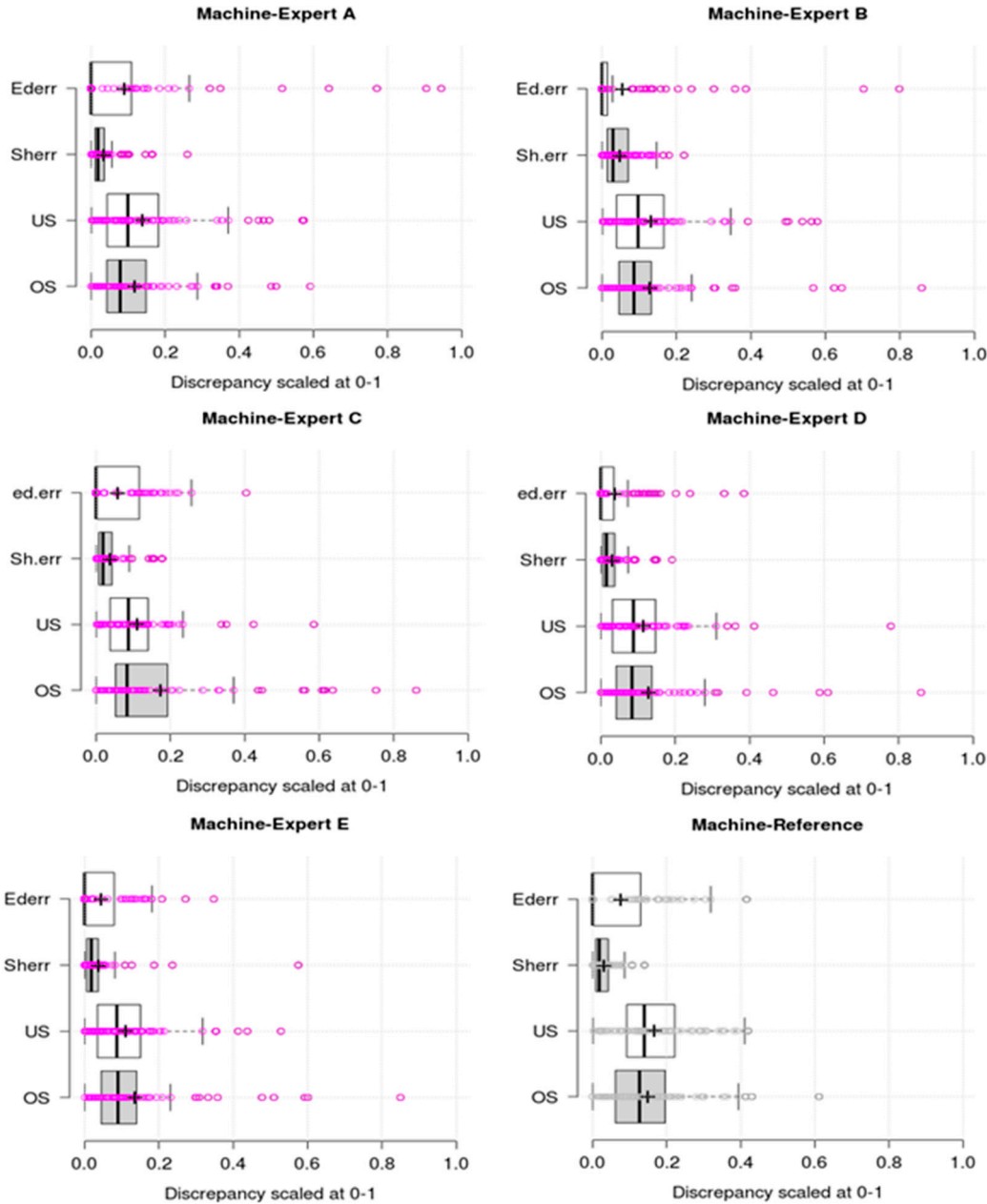

**Figure 12.** Discrepancies between automated boundaries and manually digitised parcels.

Table 3 presents the global error for each metric considered: over-segmentation, under-segmentation, shape and edge shifting. Since an optimum and error-free segmentation where OSerr, USerr, SHerr and EDerr equal 0 is the ideal case, and rare to have, we can simply define an error tolerance range within which extracted parcels are maintained as acceptable.

**Table 3.** Overall detection error by machine against humans.

|  | Machine | Expert A | Expert B | Expert C | Expert D | Expert E |
|---|---|---|---|---|---|---|
| OS | 0.15 | 0.12 | 0.13 | 0.11 | 0.11 | 0.12 |
| US | 0.17 | 0.14 | 0.13 | 0.20 | 0.15 | 0.15 |
| SH.err | 0.03 | 0.02 | 0.05 | 0.03 | 0.03 | 0.02 |
| ED.err (buffer = 4 m) | 0.07 | 0.02 | 0.06 | 0.03 | 0.03 | 0.02 |
| NSR | 0.063 | 0.049 | 0.069 | 0.108 | 0.020 | 0.059 |
| FP | 14.74% | 12.5% | 9.57% | 12.22% | 12.12% | 13.68 |

**Table 3.** *Cont.*

|  | Machine | Expert A | Expert B | Expert C | Expert D | Expert E |
|---|---|---|---|---|---|---|
| FN | 14.85% | 15.84% | 16.83 | 25.74% | 13.86% | 19.80% |
| Correctness | 47.4% | 76% | 67% | 77.8% | 77.8% | 72.6% |
| Completeness | 45% | 73% | 63% | 70% | 77% | 69% |

Reference = 100 parcels; automation = 95 parcels; expert A = 96 parcels; expert B = 94 parcels; Expert C = 90 parcels; Expert D = 99 parcels and Expert E = 95 parcels.

## 4. Discussion

### 4.1. Manual Extraction Creates Quality Issues

In general, manual digitisation of rural parcels resulted in remarkable inconsistencies among users. Not all parcels could be extracted equally, despite having the extraction guide provided to support cadastral experts. The results show a commonality of approximately 60% in detected parcels by humans, however, 40% are perceived differently amongst the experts. The findings raise questions regarding cadastral updating: arguably if human operators update boundaries with only imagery as support, they may introduce different non-systematic errors. Machines may introduce less error during cadastral updating: the algorithms used, if not changed, will follow the same logic. Therefore, beyond the issue of higher costs and time usually associated with human users, the issue of quality repeatability should also enter the discourse.

### 4.2. Semi-Automated Is More Feasible Than Fully-Automated

The findings of this research suggest that a semi-automated rather than a fully automated approach is more applicable for cadastral boundary extraction, for ready-to-use data that can be exported as a vector file, for example, to Esri ArcGIS or other GIS platforms. Semi-automated approaches with a user-developed rule set, based on experts' ground knowledge, generate better results since it is more adapted to context than the ESP2 tool. By improved results here, we mean topologically and geometrically well-structured parcel boundaries that do not require manual post-processing and editing. For instance, knowing the setback distance, a user can extract parcels within a defined distance from specific roads or rivers. Furthermore, since it is not possible to have all parcels with the same morphological conditions, to adapt to variation in size and shape, semi-automated approaches allow for a subsequent segmentation and classification.

### 4.3. Invisible Social Boundaries: A Challenge to Both Machines and Humans

#### 4.3.1. Rural Areas Offer Promise, but Inconsistency Is Evident

While colour is the primary information contained in images with which objects are extracted and separated [63], individual parcels are not reflected by different colours. In addition, parcel boundaries shown on images are often social constructs making it rather challenging to extract them. Some parcel boundaries are visible on imagery, whereas others are invisible and cannot be detected.

In our study, the separability of rural parcels was influenced by the extractability of features marking boundaries, parcel size and shape. Most of the parcels in rural sites were marked consistently by visible ditches. Ditches were extractable by the machine as separate elongated narrow strips (Figure 8) or otherwise it would be difficult to separate two parcels with the same texture. In eCognition, such elongated features like ditches are characterised by very low elliptic fit values and or being very highly asymmetric. The ditches being extractable as separate entities from parcels facilitate separating parcels from their neighbours. Experiments show that a parcel's layout, size and fragmentation affect the extraction of boundaries. Regarding shape and size of parcels, it could be observed that having regular shaped parcels eased the automation whereas highly fragmented parcels prompted omission and commission errors, due to variation in shapes and size of parcels. As was experienced, when

classifying segments with shape indexes like rectangular fit, shape index, border index, elliptic fit and compactness, the over-segmentation error was likely. To avoid this error, a parcel area threshold value was defined that would remove small (likely committed) parcels from classification. Furthermore, screening small polygons to prevent over-segmentation resulted in under-segmentation errors as there exist very tiny plots reflecting the level of land pressure in the country. Not only extracting highly fragmented parcels was a challenge for automated approaches, but also to humans. Some of the hypothesised boundaries of human experts could not necessarily match references parcels. This means that the physical line is not enough to define a boundary. This leads to human subjectivity, because of individual differences in image interpretation [64], in parcel delineation. Inconsistencies, where it is likely for one human expert not to produce the same parcels nor uniformly digitise the same boundaries repeatedly, present a weakening feature of image-based delineation.

### 4.3.2. Urban Areas Surprisingly More Challenging

Major challenges were encountered in the urban area, owing to higher heterogeneity and diversity with respect to form, size, layout and material constitution of urban structures. For instance, one roof surface may display varying spectral signatures, making it very difficult for automated building extraction. In the studied case, buildings roofs are mostly hip and valley, and prone to spectral reflectance variation. In fact, depending on the position of the sun at the time of image acquisition, some parts of the roof are not illuminated which affects the extraction, raising requirement concerns over the quality of the imagery required for cadastral mapping purposes. Not only roofs, but also the material composition of fences and marking of plot boundaries varies, making it difficult for parcel extraction. Fences are very relatively and typically narrow objects, hard to detect with a 0.5-m resolution image. In some cases, building roofs, the fences marking parcels, building shadows and gardens, had almost the same spectral signatures, making it almost impossible to separate these features.

Generally, from our experiments, it can be assumed that the extraction of buildings and urban plot boundaries, using spectral information of roofs and fences is challenging due the complexity of urban fabric and the quality of the remotely sensed data used by the machine, as opposed to humans. Looking at the image, buildings and fences are very able to be identified with eyes, and we can also see good results from digitisation by experts. Counter-intuitive results obtained from automation confirm observations made in Reference [48].

### 4.4. Still Areas of Strengths and Weakness for Both Humans and Machines

From the comparison of manually digitised boundaries against automatically generated rural parcel boundaries (Figure 11), the most striking observation is the likeness of the degree of deviation of automatically and manually extracted parcel shapes from the real (reference) parcel shapes. This demonstrates that the deviation of automated parcel shapes from manually digitised parcel shapes were too small. Results in Figure 12 were in agreement with those in Figure 11, showing that nearly all extracted parcel polygon areas by experts have less than 20% of their areas committed or omitted from automated parcels polygon areas. In general, human operators were geometrically more precise compared to machine algorithms when drawing and reproducing parcel geometries from images, but the machine's performance is auspicious in the rural context. On the contrary, in urban areas, humans outperformed automation. In fact, automated parcels and building outlines were topologically and geometrically poorly structured and not comparable to manually digitised parcels and building outlines.

### 4.5. Corroboration with Previous Studies

Our study findings are in agreement with previous studies [16,35] where obtained automation performance in rural areas was 24–65%. In contrast to findings by [50,65,66], however, automation performance was lower due to the focus on geometric precision rather than thematic accuracy. Unlike previous studies, except the study in Reference [16], this study focused on automated extraction of

whole-parcel boundaries. Here, the importance would not be to consider only higher automation rates, but also more emphasis on providing information that fit with acceptable cadastral standards. According to [35,49], even with automation performance, 30–50% will significantly reduce the cost incurred in land demarcation. Therefore, it can be concluded that the current study achieved promising results in rural areas. In urban areas, however, while an unambiguous ontology status of buildings, with shapes that are clearly detectable by humans would ease their delineation [67], results can be counterintuitive.

### *4.6. Implications for Practice and Research*

Our method for comparison can be implemented in Esri ArcGIS. It is quantitative and hence reproducible and replicable. As for implications, first, the study instils future researchers to use geometric accuracy metrics in compliance with cadastral standards.

The second implication of the study derives from the spatial quality of the obtained automation results leading to, potentially, transferability not of the rule set but the approach used. It was noted, in experimentation with the ESP2 tool, that the rule set might not be transferable instinctively. It is because the rule set includes parameter values set to fit a specific context and not the general context. Likely, the approach is designed in such a way that with small adjustments of the rule parameters pertaining to shape and size, depending on the context of the concrete case, it can be replicated in other contexts. This makes our work highly beneficial for future researchers and other case studies.

Third, from reviewed previous proponent works on the automation of cadastral boundaries extraction (as it is also for the current study as limitation) the issue of scalability emerges. Many of the inferences made in this work are based on simple case studies using smaller tiles of images which do not represent the complexity on the ground for whole-country mapping endeavours. The research problem is aligned to a real-world problem, but the presented solution primarily considers methodological matters, not the broader set of political, legal, organisational and administrative challenges. In addition, using automation in small areas might not be a wise idea, in terms of gaining critical mass and economies of scale. The implication of this is a need to apply automatic tools to a large area in simulations to real-world practices, rather than using smaller and subjectively selected areas.

Generally, beyond the requirement to understand the data, the experimentation suggests that automated extraction of cadastral boundaries, also requires knowing the social contexts that shape landholding structures in a given area. During automation, the user must integrate this contextual knowledge within the rule sets. Fully automated approaches could not be fruitfully compared with the user-developed rule set, since it limits user intervention, if not ignores it, and does not integrate expert ground knowledge.

## 5. Conclusions and Recommendation

This study compared machine driven techniques, using rule sets within OBIA, and human abilities in detecting and extracting visible cadastral boundaries from very high-resolution satellite images, in both urban and rural contexts. Our results show that automation was able to correctly extract 47.4% of visible rural parcels and achieved 45% of completeness, whereas in urban areas, it failed to generate explicit polygons owing to urban complexities and spectral reflectance confusion of cadastral features.

Machines are meant to increase human performance in production and service delivery. In the cadastral field, this will be achieved if human cadastral intelligence—knowing boundaries are social constructs and perceptible to humankind—is integrated with computational machine power to allow for the extraction of parcels to support land registration. With the obtained results in rural settings, land registration service coverage can be taken farther than is currently possible.

Despite the rigorous methods applied, the study does not claim a fully-fledged experimentation with automation tools. Thus, more studies using other tools and other case studies are required to broaden the understandings of the tools that best fit the given purpose and context. In this study, automation was applied on a relatively small area, suggesting that it could also be scaled to larger areas.

Finally, in urban areas, the study encountered limitations since the colour (i.e., spectral signatures) was the primary information contained in image data in segmentation. As an implication, the incorporation of Light Detection and Ranging (LiDAR) information may improve the obtained results and hence is suggested for the application to similar and other case studies. The overarching aim of our work on cadastral feature extraction is to generate a rough-cut cadastre that can be taken to the field and corrected, involving cadastral professionals and owners. The goal is to identify and apply user-friendly and learnable automation tools that result in precise and GIS-ready cadastral boundaries on as the largest scale possible.

**Author Contributions:** E.N. wrote the manuscript, organised and conducted field data collection and the experiments. M.K. and D.K. and R.B. contributed to the conceptual design of the experiments and reviewed the paper.

**Funding:** This research was a master study under ITC scholarship and the government of Rwanda.

**Acknowledgments:** Authors acknowledge cadastral experts, including four cadastral maintenance professionals from Rwanda Land Management Authority and one surveyor from the Organisation of Surveyors in Rwanda, for accepting to take part in this study by hypothesising and digitising parcels boundaries from images. We acknowledge the land registration department of Rwanda land management authority for providing legal boundaries. We are grateful to ITC for providing image data.

**Conflicts of Interest:** The authors declare no conflict of interest.

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
