# Peer review of "Comparing Human Versus Machine-Driven Cadastral Boundary Feature Extraction"

_remotesensing, doi:10.3390/rs11141662_

Round 1
Reviewer 1 Report
My comments mostly refer to form, presentation and discussion, and I hope the consideration of them will make the manuscript better readable and more explicit. Major comments are presented below.
SPECIFIC COMMENTS:
1) References aren’t quoted correctly (at variance with instructions to the authors). Correct it.
Abstract:
2) L12-27: The abstract doesn’t contain any information about innovation. The abstract must state the innovation in the conducted research in two or three sentences.
Keywords:
3) Please find such words which are not in the title, this way search engines of the web will find your manuscript with higher probability.
Introduction:
4) L31 and L32: Are all those references relevant? If so I would suggest that you group them according to categories or applications rather than just placing them together. In this form, there is little evidence that they have been read and there is no indication of their respective importance to the subject.
5) In context of your research see also: https://doi.org/10.1080/14498596.2017.1404500
Materials and Methods:
6) Figure 2: poor quality (including text)
7) L200: If you are using Word, please use either the Microsoft Equation Editor or the MathType add-on.
8) Where is Figure 3 about you mentioned in L191-192?
9) Figure 4: poor quality (including text)
Experimental Results:
10) L221-222: where is Figure 6?
11) L250: where is Figure 9?
References:
12) References aren’t quoted correctly (at variance with instructions to the authors). In the text, reference numbers should be placed in square brackets [ ]; for example [1], [1–3] or [1,5].
13) A comprehensive list of references must be numbered in order of appearance in the text.
Reviewer 2 Report
This manuscript compares manual and automated extraction of urban and rural features in Rwanda from HR pansharpened WorldView image. The manuscript is well organized. It needs linguistic review since I encountered several grammar and syntax errors.
Although the authors did a good job preparing the dataset and utilizing basic algorithms in this field, I have a major concern for the novelty of the manuscript and the significance of its results. Several studies attempted to use OBIA analysis for feature extraction and I don't see this manuscript adding new aspects to the body of knowledge in this area. This is evidenced in the low percentage of the correctly extracted features in the rural area and the lack of results in the urban area. I would envision considering these results as benchmark for further experimentation using different parameters or developing new algorithms to improve on the reported results.
Reviewer 3 Report
The manuscript titled “Comparing Human Versus Machine-Driven
Cadastral Boundary Feature Extraction” touches an interesting topic
and presents interesting results. Nevertheless, there are some
serious issues which would prevent the manuscript from being
published.
Starting from the major issues, I would stick to the automatic parcel extraction methodology and the methodology for computing the geometric discrepancies as well as figures 2 and 4 (figure 3 does not exist). Although the results are presented in a good manner the methodologies are not.
The methodology for the automatic parcel extraction is hardly described in the document and figure 2 is not enough for describing it. Since the methodology is not analyzed, figure 2 seems like being a wrong copy paste from another paper. Here are a lot of problems with the most important being that:
The authors have never mentioned in the text that they have used OSM (open street map layers, I suppose)
The parameters used for the multiresolution segmentation are not referenced and the different levels of the multiresolution segmentation do not seem to have been used somehow.
The authors seem to have used some of the object features and functionalities of eCognition (using fuzzy rules I guess) in order to perform the automatic boundary extraction, but these are never described in the text. It would be a good idea to create a table containing all these rules, the membership functions and the values used in a comprehensive way.
In any case, figure 2 alone (besides being rather complicated) cannot describe the workflow. The authors have to analyze the methodology and show all the rules and sub-processes that have been used for this scope.
Then the authors do the same mistake for the methodology they follow for computing the geometric discrepancies. They do not analyze the methodology, but rather place figure 4, which again is rather complicated and not easy to understand, unless the methodology is somehow described in the text.
Apart from the two major issues, which concern the two mentioned methodologies, there are some more remarks and recommendations:
There are a lot of abbreviations in the text and the figures which have not been adequately explained before their first use.
The first sentence of the introduction, which is a rather generic statement, has 14 references! I do not think this is a very good way to start an introduction.
The experiments are rather limited. It would be much better to have more study areas.
Line 264: “buffer of 4m” the authors have to explain and justify such choices
Lines 293-296: 40% inconsistencies among cadastral experts cannot be referenced as “slight inconsistencies”
Line 404: “as a preliminary step...”, this is exactly what the authors should be seeking for in the first place. When it comes to cadastral mapping, it is a common practice to create a preliminary boundary layer, in order to support the photointerpretation of the possible parcels, which will again be corrected later on before the final map creation.
p { margin-bottom: 0.25cm; line-height: 115%; background: transparent none repeat scroll 0% 0%; }Author Response
Please see the attachment.

Round 2
Reviewer 3 Report
Following the first review, the authors have significantly improved their manuscript and to my opinion the work described could be published after some minor corrections/additions. More specifically:
In 2.2.1.2: figure 2 as well as the methodology description, should become more clear as far as it concerns the merging procedure. What does “merge” mean in the first loop and how is the merging of the 9 different layers of parcels take place in the case of overlapping parcels? Are there overlapping parcels? Does the 2nd, 3rd, 4th and so on segmentation/classification take place only in the non classified areas? Please explain.
Line 195: It would be better to reference GLCM as an additional image feature, not a temporary image layer. Please explain how this feature is used. Is it just added as an additional layer which is taken into consideration during segmentation? It would also be a good idea to show a picture of it in order for the reader to understand its usefulness.
Please try to improve figure 4. Try to follow the rules of logical diagrams.
Figure 5 also needs some improvement. The circles containing “automated” and “Reference parcel dataset” are note processes and therefore should not be circles. They could be boxes outlining the relevant processes. Moreover, try to be more specific. E.g. change “automated” to “automatic boundary extraction” for instance.
At the end of the same figure “Dissolved by ID….” and “dissolved”: what does dissolved mean, how is it handled? Please add some more explanation about this graph. One line (line 291) is not enough.
Figures 6 and 7: The reference parcels of fig 6 are different than those of fig 7. Fig 7 seems to present in yellow the correct reference boundaries.
Figure 8 is of poor quality. The parcel boundaries can hardly be seen.
Figure 11: paragraph 2.2.3, which explains the abbreviations, is quite far from the figure. Add explanations for Ed, SH, US, OS in the title.
